# NeuroSchedule: A Novel Effective GNN-based Scheduling Method for High-level Synthesis

**Jun Zeng**[*]
Tsinghua University

**Mingyang Kou**[*]
Tsinghua University

**Hailong Yao**[†]
Tsinghua University

**Xu-Cheng Yin**
University of Science and Technology Beijing

**Haili Wang**
Hercules Microelectronics Co., Ltd

## Abstract

High-level synthesis (HLS) is widely used for transferring behavior-level specifications into circuit-level implementations. As a critical step in HLS, scheduling arranges the execution order of operations for enhanced performance. However, existing scheduling methods suffer from either exponential runtime or poor quality of solutions.

This paper proposes an efficient and effective GNN-based scheduling method called NeuroSchedule, with both fast runtime and enhanced solution quality. Major features are as follows: (1) The learning problem for HLS scheduling is formulated for the first time, and a new machine learning framework is proposed. (2) Pre-training models are adopted to further enhance the scalability for various scheduling problems with different settings. Experimental results show that NeuroSchedule obtains near-optimal solutions while achieving more than **50,000**× improvement in runtime compared with the ILP-based scheduling method. At the same time, NeuroSchedule improves the scheduling results by 6.10% on average compared with state-of-the-art entropy-directed method. To the best of our knowledge, this is the first GNN-based scheduling method for HLS.

## 1 Introduction

Recently, FPGA has been playing an important role in accelerating computations from the cloud to the edge [1, 2, 3, 4, 5]. As a reconfigurable computing device, FPGA gains increasing popularity for its high performance and great energy efficiency. However, developers are required to write programs using the hardware description language (HDL) to implement applications in FPGAs, which is time-consuming and error-prone. Therefore, the utilization of FPGA's enormous computing power is limited by the cumbersome developing process.

As a result, the concept of high-level synthesis (HLS) is proposed in the EDA community to speed up the developing process [6, 7, 8]. HLS automatically transfers high-level specifications (written in high-level languages like C/C++) to behavior-level implementations (written in HDLs like Verilog/VHDL). Figure 1 presents the whole flow of HLS. As shown in the figure, HLS mainly comprises three parts: compilation, synthesis, and generation. For compilation, high-level languages are transferred to Control Data Flow Graphs (CDFGs) with the assistance of modern compilers like GCC or LLVM. During the synthesis process, hardware resources (e.g. functional units, storage, and wires) are first allocated. According to the allocated hardware resources, the operations in CDFG (such as ADD and MUL) are scheduled to certain time steps. After scheduling, the allocated hardware resources are

---

[*]These authors contributed equally to this work.

[†]Corresponding author: Hailong Yao (hailongyao@tsinghua.edu.cn).

36th Conference on Neural Information Processing Systems (NeurIPS 2022).

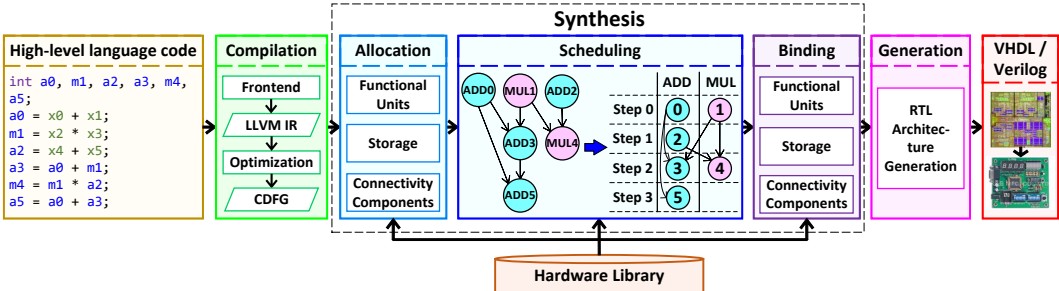

Figure 1: High-level synthesis flow.

bound to corresponding operations and time steps. After synthesis, the generation process generates corresponding HDLs according to the synthesized results.

Scheduling is one of the most critical stages in the HLS process, which arranges the execution order of operations, and thus determines the performance of generated circuits. As depicted in Figure 1, according to the allocated hardware resources and the dependencies between operations, the operations in CDFG are arranged in different time steps. For example, during the scheduling process, operation ADD0 is scheduled to step 0 and bound to the adder. It is noteworthy that only one adder is allocated to the scheduling task. So operation ADD2 could not be scheduled to step 0 same as operation ADD0. Therefore, determining the scheduling priority for the operations is the key objective of the scheduling problem. However, the scheduling problem has been proved to be NP-hard [9]. And existing algorithms suffer from either unbearable long runtime or poor quality of solutions. Integer linear programming (ILP) formulation has been proposed to solve the scheduling problem optimally [10]. However, the exponential runtime of ILP is unbearable. Therefore, various heuristic algorithms are proposed to obtain sub-optimal solutions within acceptable runtime. List scheduling is one of the most famous heuristic algorithms [11]. List scheduling stores ready operations in a list and arranges them according to a heuristically defined priority function. Based on the list scheduling algorithm, force-directed scheduling algorithm is proposed [12], which builds a force-distribution graph for representing the priority function. Inspired by the force-directed scheduling algorithm, Entropy-Directed Scheduling (EDS) algorithm [13] replaces the priority function with the entropy of CDFGs to be scheduled. The utilization of the entropy function not only speeds up the scheduling process, but also improves the quality of solutions. Different from the algorithms in the list scheduling family, an algorithm based on the system of difference constraints (SDC) is proposed to speed up the ILP-based algorithm. SDC-based algorithm transfers the scheduling constraints into a system of difference constraints using heuristics, which greatly reduces the number of equations to be solved. However, the SDC-based algorithm cannot guarantee the optimal solution. Therefore, the aforementioned heuristic algorithms all trade quality of solutions for runtime, which leaves a large optimization space to be explored.

On the one hand, the complicated dependencies of operations in the CDFG bring great challenges to the scheduling process. On the other hand, Graph Neural Networks (GNNs) [14, 15, 16, 17] have great representational power of complicated graphs, which is suitable for extracting complicated relations between operations in CDFGs. Therefore, a natural idea is to adopt GNNs to tackle the scheduling problem for both fast runtime and high-quality solutions. This paper proposes the first GNN-based scheduling algorithm called NeuroSchedule. Inspired by algorithms in the list scheduling family, NeuroSchedule builds a GNN model to predict the priorities of operations in the CDFG. The predicted priorities are then used to arrange the operations in CDFG to obtain the scheduling results. Major contributions of this paper are summarized as follows:

- A learning problem for the HLS scheduling is first formulated, and then a GNN-based learning framework is proposed to effectively and efficiently solve the problem.
- Pre-training models are adopted to further enhance the scalability for different scheduling problems with different settings.
- Experimental results indicate that NeuroSchedule obtains near-optimal solutions while achieving more than **50,000**× speedup compared with the ILP-based algorithm [10]. At the same time, NeuroSchedule improves the scheduling results by 6.10% on average compared with the state-of-the-art EDS algorithm [13].

The rest of this paper is organized as follows. Section 2 gives a motivating example for the GNN-based scheduling method. Section 3 gives preliminaries and the formulation of the GNN-based

scheduling problem. The proposed machine learning framework is presented in Section 4. Section 5.2 introduces NeuroSchedule, as well as the pre-training models adopted for enhancing the scalability. Section 6 presents the experimental results. Finally, conclusion is drawn in Section 7.

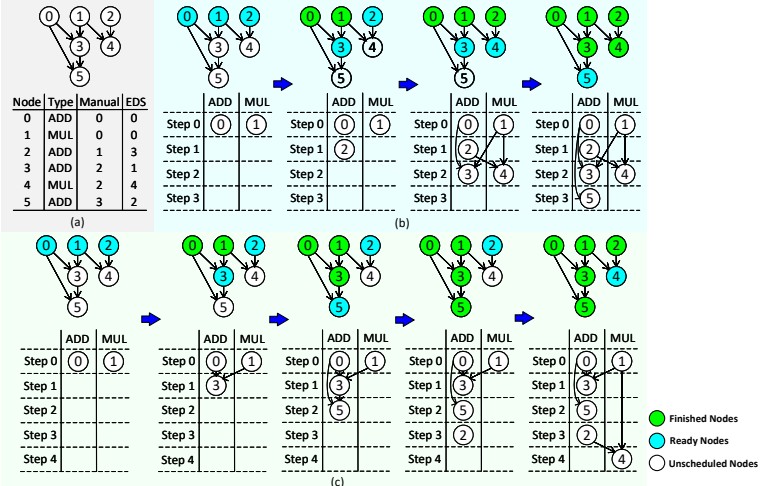

Figure 2: A motivating Example. (a) CDFG and the priority of nodes. (b) Manually designed list scheduling result (4 cycles). (c) Entropy-directed scheduling result (5 cycles).

## 2 Motivation

As introduced in Section 1, the list scheduling algorithm stores ready operations in a list and schedules them according to a heuristically defined priority function. Here, a ready operation means all of its predecessing operations are finished and the operation could be scheduled for execution immediately. For example, in Figure 2(a), node 3 is ready when node 0 and node 1 are finished. The process of list scheduling could be stated as follows. First, the ready operations are stored in a list. Next, the operation with the highest priority in the list is picked to be scheduled to available hardware resources (e.g. adders and multipliers) for execution. After the operation is finished, the newly generated ready operations are stored in the list. The above process is iterated until all the operations are successfully scheduled. From the list scheduling process, defining precise priority functions is crucial to obtain good scheduling results.

A motivating example is given in Figure 2. Figure 2(a) presents the CDFG to be scheduled, which consists of 6 operations with 2 types (ADD and MUL). And the available hardware resources for scheduling are an adder (for ADD) and a multiplier (for MUL). Figure 2(c) depicts the entropy-directed scheduling process, which takes the entropy function for the priority computation. In the figure, ready operations are colored blue and finished operations are colored green. Since the entropy priority of node 3 is higher than node 2, node 3 is scheduled before node 2. However, scheduling node 3 first delays the scheduling of node 4, which makes the multiplier idle for 3 cycles (i.e. time steps). Consequently, the final scheduling result is not optimal. In Figure 2(b), we assume there is an optimal manually designed priority function. Based on the optimal priority function, node 2 is scheduled before node 3, and thus the hardware resources are fully utilized. Therefore, the scheduling algorithm based on the manual function gives the optimal solution.

The success of the manually designed function motivates us to build a machine learning framework, which learns the priority function automatically. GNNs are an effective framework for representation learning of graphs, which follow a neighborhood aggregation scheme. In GNNs, representation vectors of nodes are computed by recursively aggregating and transforming representation vectors of their neighboring nodes [14]. Therefore, GNNs are suitable to learn the CDFG embeddings for obtaining the priority function. As a result, a GNN-based machine learning framework is proposed to learn the priority function. Experiments verify the effectiveness of the proposed scheduling method based on GNN.

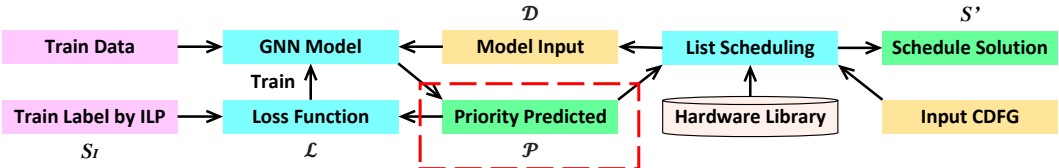

Figure 3: Framework for NeuroSchedule.

## 3 Preliminaries and Problem Formulation

**CDFG.** In high-level synthesis, the program in high-level language is first translated into CDFG, a graph-based representation defined as follows:

**Definition 1** *A CDFG $G = (V, E)$ is a directed graph with node set $V$ and edge set $E$. Each node $v \in V$ represents an operation, e.g. add or load. Each directed edge $e = (v_i, v_j) \in E, i, j \in 1, ..., |V|$ represents a dependency relationship between operation $v_i$ and $v_j$, i.e., operation $v_j$ must be executed after the execution of operation $v_i$.*

**ASAP and ALAP.** The As-Soon-As-Possible (ASAP) algorithm and the As-Late-As-Possible (ALAP) algorithm give the information about node flexibility during the scheduling process. For node $v_i \in V$, the result of ASAP and ALAP can be formulated as follows:

$$ASAP(v_i) = \begin{cases} \max ASAP(v_j) + 1 & \exists (v_j, v_i) \in E, v_j \in V \\ 0 & \nexists (v_j, v_i) \in E, v_j \in V \end{cases} \tag{1}$$

$$ALAP(v_i) = \begin{cases} \min ALAP(v_j) - 1 & \exists (v_i, v_j) \in E, v_j \in V \\ \max_{v_k \in V} ASAP(v_k) & \nexists (v_i, v_j) \in E, v_j \in V \end{cases} \tag{2}$$

**Resource-constrained scheduling problem.** Based on the CDFG, we have to solve the resource-constrained scheduling problem: given the number of resources, find the scheduling solution with the minimum number of execution cycles, while satisfying the given set of scheduling and resource constraints. Given a functional unit set $U$, where $u_i \in U$ represents a functional unit, such as an adder. Denote $d_{u_i}$ as the cycle delay of functional unit $u_i$. Given a CDFG $G = (V, E)$, and let $n = |V|$, a *schedule* can be defined as $S = (t_1, t_2, ..., t_n, f_1, f_2, ..., f_n)$, where $\forall i \in 1, 2, ..., n$, (1) $t_i \in \mathbb{N}^*$, denoting the start cycle of operation $v_i \in V$, (2) $f_i \in U$, denoting the functional unit used by operation $v_i$. Let $L = \max_{i \in 1, ..., n} t_i + d_{f_i} - 1$ denote the total number of execution cycles. The resource-constrained scheduling problem minimizes $L$, which can be defined as follows:

**Definition 2** *Find an optimal schedule $S' = (t'_1, t'_2, ..., t'_n, f'_1, f'_2, ..., f'_n)$, such that:*

*a)* $\forall e = (v_i, v_j) \in E, i, j \in 1, 2, ..., n, t'_i + d_{f'_i} \leq t'_j$

*b)* $\forall i, j \in 1, 2, ..., n, i \neq j, f'_i = f'_j \implies |t'_i - t'_j| \geq d_{f'_i}$

*c)* $L = \max_{i \in 1, ..., n} t'_i + d_{f'_i} - 1$ *is minimized.*

**List scheduling.** As introduced in Section 2, list scheduling is an effective scheduling algorithm. It divides the node set $V$ into three sets, finished nodes $V_F$, ready nodes $V_R$, and unscheduled nodes $V_U$. Given the node priority function $\mathcal{F}$, where $\mathcal{F}(v_i) < \mathcal{F}(v_j), v_i, v_j \in V_R$ indicates that operation $v_i$ has a higher priority than operation $v_j$. At each step, list scheduling algorithm chooses the node $v_h \in V_R$ to schedule, where $\mathcal{F}(v_h) = \min_{v_r \in V_R} \mathcal{F}(v_r)$. A better priority function in the list scheduling algorithm corresponds to a better scheduling solution. As the ILP-based scheduling algorithm can get the optimal scheduling results, the scheduling result $S_I = (t_{I1}, t_{I2}, ..., t_{In}, ...)$ is a good candidate for the priority function, where $\mathcal{F}(v_x) = t_{Ix}, v_x \in V$. However, for large scheduling problems, the runtime of the ILP-based algorithm is not acceptable. Therefore, we propose the GNN model to predict the priority function.

**Neural Network Model.** Denoting the input data as $\mathcal{D}$, our model can be formulated as:

$$\mathcal{P}(i) = MLP(GIN\_EMBED(\mathcal{D}_i)) \tag{3}$$

Our loss function can be formulated as:

$$\mathcal{L} = \mathcal{L}(\mathcal{P}, S_I) \tag{4}$$

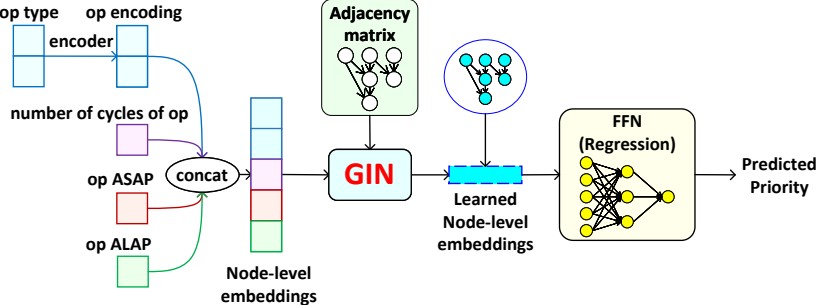

Figure 4: Model Architecture.

Details of our loss function will be introduced in Section 4. Figure 3 shows the framework of our NeuroSchedule algorithm. The ILP-based scheduling algorithm is adopted to generate the training labels for our GNN model. The output of the GNN model, i.e. $\mathcal{P}$, is used by the list scheduling algorithm as the priority function, which obtains significantly enhanced scheduling solution.

# 4   Model Design

A machine learning framework is proposed to learn the priority function in the list scheduling algorithm. In the proposed framework, the CDFG is taken as input, and a GNN model is trained to predict the priorities of the operations. This section gives the model input, the model architecture, and the selection of loss function.

## 4.1   Model Input

The input of a GNN model includes an adjacency matrix representing graph connections and a tensor representing node features. Different node features will be selected for different problems. In the scheduling problem, the node features consist of operation type, number of cycles of the operation, along with the above-defined ASAP and ALAP.

**Operation's type and number of cycles.**  As the operation type is categorical and discrete, inspired by the graph-based program representation [18], the operation type is encoded by one-hot encoding. For the number of cycles of an operation, the cycle number is directly used in the encoding. For example, if there are two types of operations (MUL, ADD) and the cycle number of MUL operation is 1, the node feature of MUL operation could be represented as [1, 0, 1]. Here, "10" from the first "1" and second "0" in the feature denotes MUL.

**Operation's ASAP and ALAP.** Except for operation's type and number of cycles, ASAP and ALAP are also adopted in the node features. As introduced in Section 3, ASAP and ALAP represent the earliest step and the latest step that an operation could be scheduled, respectively. The ASAP and ALAP of an operation indicate the operation's flexibility during scheduling. For example, the ASAP of node ADD2 in Figure 5 is step 0 and the ALAP is step 1, which means node ADD2 could be scheduled between step 0 and step 1. Different from node ADD2, the ASAP and ALAP of node ADD3 in Figure 5 are both step 1, which means node ADD3 could only be scheduled to step 1. The reason why ASAP and ALAP are adopted in the node features is that the flexibility of an operation affects its priority during scheduling. Operations with low flexibility during scheduling tend to be the bottlenecks [13], which should be scheduled with high priority to fully utilize the hardware resources.

Finally, the operation's type, operation's number of cycles, along with normalized ASAP and ALAP, are concatenated as a tensor representing node features. For example, there are two operations in

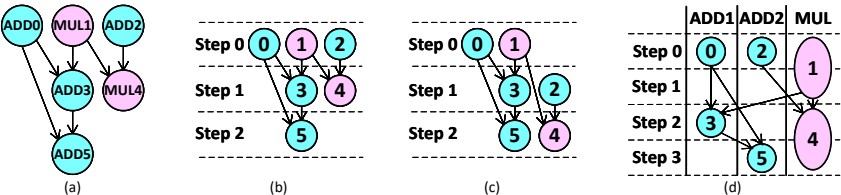

Figure 5: An example for ASAP, ALAP, and multi-cycle operations. (a) The CDFG. (b) The result of ASAP. (c) The result of ALAP. (d) A special scheduling example with multi-cycle operations.

Figure 5, MUL and ADD, and the number of cycles of operations MUL and ADD are both 1. So node ADD3 could be encoded as [0, 1, 1, 0.5, 0.5]. It is noteworthy that the ASAP and ALAP of an operation are normalized according to the maximum ASAP value among all the operations. The normalization is conducted because the number of operations in the CDFG varies such that using the absolute ASAP and ALAP values will bring noises.

## 4.2 Model Architecture

Figure 4 presents the architecture of the proposed model. As shown in the figure, the proposed model is composed of a GNN model and a feed-forward network (FFN) model. The GNN model is adopted to learn the embedding of each operation from the operation features and graph structure, while the FFN model is utilized to conduct regression tasks for predicting operations' priorities. The workflow of the model is presented as follows. First, operation features including the operation's type, operation's number of cycles, along with normalized ASAP and ALAP are concatenated into a feature tensor as introduced in Section 4.1. Next, a GNN model is adopted to learn operation embedding from the feature tensor and graph structure. Graph Isomorphism Network (GIN) is used as the GNN model for its strong power of learning embeddings from graph structures [14]. Finally, the learned operation embedding is fed into the FFN model for predicting the operations' priorities. In the FFN model, there are three fully-connected layers, and ReLU is adopted as the activation function [19].

## 4.3 Training Settings

As depicted in Figure 4, the proposed model outputs the operations' priorities by regression. To effectively train the proposed model, a dedicated training pipeline is proposed. The proposed training pipeline mainly focuses on two questions: 1. How to acquire the regression labels; 2. How to select the training objective function (a.k.a loss function).

**Training label.** As formulated in Section 3, training the priority prediction task requires a collection of data $\mathcal{D} = \{g_i, p_i, c_i\}$, where $g_i$ denotes CDFG's graph structure, $p_i$ denotes operation properties (such as ASAP and ALAP), and $c_i$ denotes the training label. In the proposed training pipeline, the ILP-based method is used to generate the regression labels [10]. The details are presented as follows. The ILP-based method gives the optimal solution, which arranges CDFG's operations into suitable time steps. Obviously, operations arranged in later time steps have lower priorities than those arranged in earlier time steps. Therefore, in the proposed training pipeline, the time step obtained by the ILP-based method (i.e. $t_{Ii}$ in Section 3) is taken as the regression label $c_i$. As mentioned in Section 4.1, the time step of an operation is also normalized based on the maximum time step, which reduces the impact of noises. Figure 2 gives an example. In the manually designed priority function, the time step could be solved by the ILP-based method. As shown in the figure, the ILP-solved time steps $[0, 0, 1, 2, 2, 3]$ are assigned to corresponding operations. After normalization, the ILP-solved time steps are computed as $[0, 0, 0.33, 0.66, 0.66, 1]$.

**Training objective function.** The mean square error (MSE) function is commonly adopted for training a regression task. Given dataset $\mathcal{D} = \{g_i, p_i, c_i\}$, the output of our GNN model has the form $\mathcal{P}(i) = \mathcal{P}(g_i, p_i)$. The MSE function $\sum_i (\mathcal{P}(i) - c_i)^2$ guides the model to accurately predict the operations' priorities. However, the scheduling task cares more about the operations' relative order rather than the absolute priority values of the operations. Moreover, it takes more effort for training a model to predict the absolute value than to predict the rank. Therefore, inspired by AutoTVM [20], the following rank loss function is adopted to train the model for predicting the rank of all operations [21]:

$$\mathcal{L} = \sum_{i,j} \log(1 + e^{-\operatorname{sign}(c_i - c_j)(\mathcal{P}(i) - \mathcal{P}(j))}). \tag{5}$$

During training, the above rank loss function keeps the relative order of operations' priorities with reduced training efforts.

# 5 NeuroSchedule

## 5.1 Pre-training for Different Scheduling Settings

As introduced in Section 4.1, the input of our model consists of the operation's type and operation's number of cycles, along with normalized ASAP and ALAP. It works well when the numbers of

---
**Algorithm 1** NeuroSchedule algorithm.
---
**Input:** Trained network model $\mathcal{P}$, CDFG $G = (V, E)$, functional unit set $U$, delay array $d$
**Output:** Generated schedule $S_{\mathcal{P}}$
 1: $V_F \leftarrow \phi, V_R \leftarrow \phi, V_U \leftarrow V, S_{\mathcal{P}} \leftarrow \phi, Time \leftarrow 0$
 2: $\mathcal{D} \leftarrow encodeData(G)$
 3: $\mathcal{F} \leftarrow runModel(\mathcal{P}, \mathcal{D})$
 4: **for** $v_i$ in $V$ **do**
 5:      **if** $v_i.inDegree() = 0$ **then**
 6:          $V_R \leftarrow V_R \cup v_i, V_U \leftarrow V_U \setminus v_i$
 7: **while** $|V_F| < |V|$ **do**
 8:      **for** $u_j$ in $U$ **do**
 9:          $M \leftarrow MAX\_FLOAT, v_{chosen} \leftarrow \phi$
10:          **for** $v_i$ in $V_R$ **do**
11:              **if** $\mathcal{F}(v_i) < M$ **and** $v_i.canUse(u_j)$ **then**
12:                  $M \leftarrow \mathcal{F}(v_i), v_{chosen} \leftarrow v_i$
13:          **if** $v_{chosen} = \phi$ **or** $u_j.unavailable(Time)$ **then**
14:              **continue**
15:          $S_{\mathcal{P}}.insert(v_{chosen}, u_j, Time)$
16:          $u_j.setUnavailable(Time, Time + d_{u_j} - 1)$
17:          $V_R \leftarrow V_R \setminus v_{chosen}, V_F \leftarrow V_F \cup v_{chosen}$
18:          **for** $v_i$ in $V_R$ **do**
19:              **if** $(v_{chosen}, v_i) \in E$ **then**
20:                  $E \leftarrow E \setminus (v_{chosen}, v_i)$
21:                  **if** $v_i \in V_U$ **and** $v_i.inDegree() = 0$ **then**
22:                      $V_R \leftarrow V_R \cup v_i, V_U \leftarrow V_U \setminus v_i$
23:      $Time \leftarrow Time + 1$
---

functional units are fixed. However, there are various scheduling settings in real HLS cases including: (1) different operation types, (2) different number of functional units for each type, and (3) different number of cycles for different operation types. Figure 5(d) gives a scheduling example with 2 adders, 1 multiplexer, and multi-cycle operations, i.e., the MUL operation takes 2 cycles. Therefore, for scheduling problems with different settings, specified GNN models are required to get valid solutions. However, it is infeasible to train a GNN model for each scheduling setting due to the unbearable training efforts. Therefore, we build a dataset of CDFGs with different scheduling settings, and pre-train our model with the dataset. To specify the scheduling setting, we add the number of the corresponding functional units into the node features of each operation. While dealing with a specific scheduling setting, we fine-tune the model using the randomly generated data from the setting. Compared with direct training, the fine-tuning takes fewer efforts with enhanced scheduling solutions.

## 5.2 NeuroSchedule Algorithm

Algorithm 1 gives an overview of the proposed NeuroSchedule algorithm, which is based on the traditional list scheduling algorithm and our neural network models. The symbols in Algorithm 1 are defined in Section 3. We use trained network model $\mathcal{P}$ as priority function $\mathcal{F}$ (line 3), and run the loop until each node in the CDFG is successfully scheduled (line 7). In each cycle, we try to schedule every available functional unit using the node $v_{chosen}$ with the highest priority. After the node $v_{chosen}$ is selected from the ready set $V_R$, we update the schedule $S_{\mathcal{P}}$, and the sets $V_R$ and $V_F$ (line 16). Then we delete all the out edges $(v_i, v_{chosen})$ of $v_{chosen}$ (line 20). If the in-degree of $v_i$ equals 0, we add $v_i$ into the ready set $V_R$. Finally, the scheduling solution is obtained in $S_{\mathcal{P}}$.

## 6 Experiments

The proposed GNN-based scheduler is implemented in Python (version 3.9.12), and the GNN model is designed and trained with Pytorch [22] (version 1.8.0) and pyG [23] (version 2.0.4). To train the GNN model, we build a dataset including 50,000 CDFGs. The operations in the CDFGs are annotated with the priorities solved by the ILP-based method (as introduced in Section 4.3). For efficiency, Gurobi [24] is adopted as the ILP solver. Details of dataset preparation are presented in supplementary materials. We conduct all the experiments on a Ubuntu 20.04 LTS Linux Server with a CPU (Intel(R) Xeon(R) Gold 5218 CPU@2.30GHz) and a GPU (NVIDIA Tesla V100). The experiments are conducted to answer the following questions:

Table 1: Results of NeuroSchedule.

| Testcase | | | CDFG size | | NeuroSchedule | | Entropy-directed | | ILP-based | | Result | |
|---|---|---|---|---|---|---|---|---|---|---|---|---|
| Dataset | Name | Basic Block | Node | Edge | Cycle | Time | Cycle | Time | Cycle | Time | $\delta^{1}$ | Imp.[2] |
| CHStone | adpcm | main | 78 | 174 | **52** | 0.019 | 54 | 0.004 | 52 | 2.842 | 0.00% | 3.70% |
| CHStone | aes | ARK_InvCol | 79 | 192 | **77** | 0.013 | **77** | 0.005 | 77 | 2.430 | 0.00% | 0.00% |
| CHStone | blowfish | BF_set_key | 147 | 456 | **80** | 0.107 | 81 | 0.010 | 80 | 14.67 | 0.00% | 1.23% |
| CHStone | dfadd | R&PFloat64 | 92 | 238 | 49 | 0.013 | 50 | 0.005 | 48 | 3.570 | 2.04% | 2.00% |
| CHStone | dfdiv | float64_div | 433 | 1252 | **235** | 0.060 | 256 | 0.027 | N/A | $>10^{4}$ | / | 8.20% |
| CHStone | dfmul | float64_mul | 339 | 914 | **192** | 0.115 | 207 | 0.019 | N/A | $>10^{4}$ | / | 7.25% |
| CHStone | dfsin | float64_div | 392 | 1187 | **203** | 0.063 | 224 | 0.026 | N/A | $>10^{4}$ | / | 9.38% |
| CHStone | gsm | main | 68 | 148 | 41 | 0.029 | 41 | 0.003 | 40 | 2.121 | 2.44% | 0.00% |
| CHStone | jpeg | write4Blocks | 213 | 692 | **127** | 0.022 | 132 | 0.014 | N/A | $>10^{4}$ | / | 3.79% |
| CHStone | mips | main | 364 | 1433 | **200** | 0.038 | **200** | 0.025 | 200 | 2207 | 0.00% | 0.00% |
| CHStone | motion | mt_Vectors | 25 | 59 | **14** | 0.010 | 16 | 0.003 | 14 | 0.509 | 0.00% | 12.5% |
| CHStone | sha | final | 79 | 202 | **49** | 0.012 | 55 | 0.004 | 49 | 4.141 | 0.00% | 10.9% |
| MiBench | cosine1 | cosine1 | 66 | 76 | **43** | 0.025 | 44 | 0.006 | 43 | 41.95 | 0.00% | 2.27% |
| MiBench | cosine2 | cosine2 | 82 | 91 | **57** | 0.100 | 58 | 0.003 | 57 | 11.07 | 0.00% | 1.72% |
| MiBench | fir1 | fir1 | 44 | 43 | **34** | 0.098 | 34 | 0.002 | 34 | 1.244 | 0.00% | 0.00% |
| MiBench | fir2 | fir2 | 40 | 39 | **31** | 0.028 | 32 | 0.002 | 31 | 1.478 | 0.00% | 3.13% |
| MiBench | idctcol | dfg_3 | 114 | 164 | **69** | 0.103 | **69** | 0.004 | 69 | 16.22 | 0.00% | 0.00% |
| MiBench | write | bmp_7 | 88 | 202 | **68** | 0.097 | 69 | 0.004 | 68 | 5.759 | 0.00% | 1.45% |
| Total | | | 2761 | 7448 | **1621** | 0.952 | **1699** | 0.167 | / | $>5\times10^{4}$ | | |
| Optimal Rate | | | | | **88.89%** | | **16.67%** | | N/A | | | |
| Average Improvement | | | | | | | | | | | **0.30%** | **6.10%**[3] |

[1] $\delta$ = (NeuroSchedule - ILP) / ILP.
[2] Improvement = (EDS - NeuroSchedule) / EDS.
[3] Average Improvement = ($\Sigma$ Cycle_NeuroSchedule - $\Sigma$ Cycle_ILP) / $\Sigma$ Cycle_ILP. The optimal solutions (Imp. = 0.00%) are not counted in.

**Q1:** Could NeuroSchedule obtain better solutions than the entropy-directed algorithm?
**Q2:** Could NeuroSchedule obtain optimal solutions with respect to the ILP-based method?
**Q3:** How fast is the proposed algorithm compared against the ILP-based method?
**Q4:** How do the pre-training methods work in enhancing the algorithm's scalability?
**Q5:** What are the differences between different loss functions?

## 6.1 Quality of Solutions

To answer **Q1**, we evaluate the performance of NeuroSchedule against state-of-the-art entropy-directed scheduling algorithm [13] and ILP-based method [10] using the benchmarks from CHStone [25] and MiBench [26]. The experimental results are presented in Table 1. As shown in the table, NeuroSchedule improves the scheduling results by 6.10% on average compared with EDS. Moreover, compared with the ILP-based method, NeuroSchedule obtains optimal solutions in 88.89% of the benchmarks and near-optimal solutions in the rest, while EDS can only obtain 16.67% optimal solutions in all benchmarks.

To answer **Q2**, we synthesize a benchmark suite including large CDFGs with complex dependencies among the operations. The entangled dependencies make it hard for scheduling algorithms to find an optimal solution. We evaluate NeruoSchedule, entropy-directed algorithm [13] and ILP-based algorithm [10] on the synthesized complex benchmark suite. The results are presented in Figure 6. Figure 6 gives the heatmaps for comparing the scheduling results among NeuroSchedule, entropy-directed algorithm, and ILP-based algorithm. As shown in Figure 6(a), NeuroSchedule obtains optimal solutions on 90% of the synthesized complex benchmarks. While in Figure 6(b), the entropy-directed algorithm fails to obtain optimal solutions on all of the benchmarks due to the complex operation dependencies. Moreover, as shown in Figure 6(c), NeuroSchedule obtains better solutions on 94% of the synthesized complex benchmarks compared with the entropy-directed algorithm.

## 6.2 Execution Time

To answer **Q3**, we evaluate the runtime of NeuroSchedule against state-of-the-art entropy-directed scheduling algorithm and ILP-based method [10] using benchmarks from CHStone [25] and MiBench [26]. The results are presented in Table 1. As shown in the table, NeuroSchedule achieves more than **50,000×** speedup compared with the ILP-based method, while obtaining near-optimal solutions. Moreover, the ILP-based method fails to solve certain benchmarks within 10,000 seconds due to the exponential runtime overhead. Besides, since NeuroSchedule needs to launch CUDA kernels to compute operation priorities, it runs a little slower than the entropy-directed algorithm.

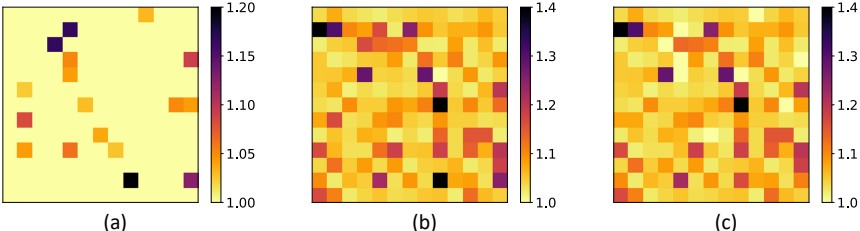

Figure 6: Evaluations on synthesized complex benchmark suite. The values in the heatmaps represent the relative scheduling results between different algorithms. (a) NeuroSchedule relative to ILP-based algorithm. (b) Entropy-directed algorithm relative to ILP-based algorithm. (c) Entropy-directed algorithm relative to NeuroSchedule.

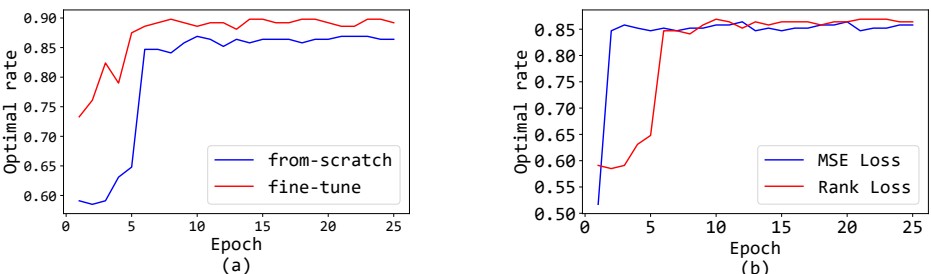

Figure 7: Optimal rate curve during training. Optimal rate indicates the number of benchmarks in the synthesized suite with optimal solutions obtained by NeuroSchedule. (a) Fine-tuning method compared against training from scratch. (b) Training with rank loss function compared against training with MSE loss function.

Nevertheless, NeuroSchedule is still fast and the difference in runtime between NeuroSchedule and entropy-directed algorithm could be ignored.

## 6.3 Pre-training and Fine-tuning

To answer **Q4**, we synthesize a small dataset that consists of 1000 CDFGs. Details of the small dataset are presented in supplementary materials. Based on the synthesized small dataset, NeuroSchedule is trained in two different ways: from scratch and fined-tuned from the pre-training model. The synthesized benchmark suite in Section 6.1 is adopted to evaluate the two models trained in different ways. Besides, the optimal rate is selected as the metric, which indicates the number of benchmarks in the synthesized suite with optimal solutions obtained by NeuroSchedule. Figure 7 depicts the optimal rate curve during training. As shown in the figure, NeuroSchedule fine-tuned from the pre-training model achieves 3% improvement in the optimal rate compared with that trained from scratch. Moreover, NeuroSchedule fine-tuned from the pre-training model achieves a better optimal rate while costing reduced training efforts compared with that trained from scratch.

## 6.4 Training Settings

To answer **Q5**, NeuroSchedule is trained with MSE loss and Rank loss [21] on the synthesized small dataset described in Section 6.3. The synthesized benchmark suite in Section 6.1 is adopted to evaluate the two models trained with different loss functions. Besides, the optimal rate is selected as the metric. Figure 7 depicts the optimal curves during training. As shown in the figure, rank loss behaves slightly better than MSE loss, which indicates that training a model for predicting the absolute value requires more effort than training a model for predicting the rank. Therefore, the rank loss is chosen as the default training objective function.

## 7 Conclusion

This paper presents NeuroSchedule, a novel and effective GNN-based scheduling method for HLS. NeuroSchedule learns to predict operation priorities using GNN models. The predicted priorities are adopted in list scheduling process. Besides, to enhance NeuroSchedule's scalability for various scheduling problems with different settings, pre-training methods are adopted. Experimental results

indicate that NeuroSchedule achieves significantly enhanced solutions compared with the state-of-the-art method. Moreover, NeuroSchedule obtains near-optimal solutions while achieving more than **50,000**$\times$ speedup compared with the ILP-based algorithm. Future work includes elaborations on the training details for better tuning, supporting operation chaining, and testing of NeuroSchedule on larger and more complicated benchmarks.

## Acknowledgments and Disclosure of Funding

This work was supported in part by the Key R&D Program of China (No.2019YFB2205002), the Key Program of NSFC (No. 62034005), the NSFC (No. 61974084), and Beijing Municipal Science & Technology Commission (No. Z191100007519015).

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
