# OpenReview forum: "NeuroSchedule: A Novel Effective GNN-based Scheduling Method for High-level Synthesis"
_NeurIPS.cc/2022/Conference — NeurIPS 2022 Accept_

### Official Review · Reviewer_PRGS · 2022-07-11

**Rating:** 6
**Confidence:** 3
**Soundness:** 3 good
**Presentation:** 2 fair
**Contribution:** 2 fair

**Summary:**

This paper implements a task scheduler for HLS based on the GNN model, and the suitable method for HLS scheduling is proposed for the first time. In hardware programming, it is necessary to convert a high-level language into a circuit-level implementation, but this step is time-consuming and yields poor quality in traditional methods. According to the characteristics of many data dependencies in the generation of CDFG during the compilation process, combined with the GNN model that can perform well on complex graphs, the proposed scheduling method can speed up the execution time of the synthesis step and improve the quality of the generated circuit-level language.

**Questions:**

1. It is not clear enough at Equations (1) & (2), considering that priority determination is the key to scheduling steps, more details need to explain ASAP/ALAP?
2. In line 138 the author indicates that the smaller value the higher priority is, but in line 140 it indicates the algorithm schedules the largest value, the description here is confusing.
3. The description of the encoding operator in line 181 is not clear, and further normalized features can reduce the impact of noises, which is not well explained in the paper.
4. In Section 5.2, it is mentioned that V_chosen on line 25 of the algorithm, but the algorithm has only 23 lines, which makes it difficult to understand.
5. What is the final time interval for different scheduling sequences in Figure 5 (b) and (c)? This supplement would give a intuitive explanation on the operation’s flexibility during scheduling.


**Limitations:**

yes

**Strengths And Weaknesses:**

Strengths
1. The paper gives a clear example to illustrate the targeted optimization algorithm, which can clearly explain the purpose of this article, and the writing logic is clear, the motivation and optimization point of this paper is clearly clarified.
2. It converts the overall priority to a relative order that is more convenient for calculation when performing operations, thereby reducing the complexity of the overall calculation.
3. The application scenario is consistent with GNN, and the priority of each operator is predicted through the GNN model, thereby reducing the execution time of the entire process.

Weaknesses
1. Some descriptions need to be more detailed. For example, when it is stated in the Introduction that operators of the same type in each step are mutually exclusive, a restrictive description of the execution should be given to avoid readers' guessing.
2. NeuroSchedule introduces GPU for calculation. The experimental results only compare the accuracy with the EDS method (in Section 6.1), emphasizing the improvement of the accuracy, and the final execution time is longer than that of EDS. The paper lacks the analysis of the benefit ratio of additional computing device.

---

> ### Author Response · Authors · 2022-08-02
> **Authors' Response to Reviewer PRGS**
>
> We thank the reviewer for your valuable comments and respond to the questions below.
>
> > It is not clear enough at Equations (1) & (2), considering that priority determination is the key to scheduling steps, more details need to explain ASAP/ALAP?
>
> In our method, the ASAP algorithm gives the time-step results to run the operations as-soon-as-possible, and the ALAP algorithm gives the time-step results to run the operations as-late-as-possible. Therefore, the vertices' flexibility can be obtained from the difference of ASAP and ALAP.
>
> In Equation (1), for vertex $v_i$, an edge $(v_j, v_i) \in E$ means that $v_i$ must be executed after the execution of $v_j$. Therefore, in ASAP algorithm, $v_i$ can be executed after all its predecessor vertices are finished. So we have $ASAP(v_i) = \max {ASAP(v_j)} + 1$. If $v_i$ has no predecessor vertices (i.e., $\nexists (v_j, v_i) \in E, v_j \in V$), it can be executed at the beginning, so $ASAP(v_i) = 0$. So we have Equation (1).
>
> In Equation (2), ASAP is computed before ALAP to obtain the depth of CDFG $G$ (i.e. $\max \limits_{v_k \in V} ASAP(v_k)$ ). According to the ALAP algorithm, for those vertices not connected to $v_i$, their ALAP values are defined as the depth of CDFG $G$ (i.e. $\max \limits_{v_k \in V} ASAP(v_k)$ ). Naturally, the operations at the bottom of the CDFG are executed as-late-as-possible. For vertex $v_i$, an edge $(v_i, v_j) \in E$ means that $v_i$ has a successor vertex $v_j$. In ALAP algorithm, $v_i$ must be finished before the execution of all the successors. So we have $ASAP(v_i) = \min {ALAP(v_j)} - 1$. So we have Equation (2).
>
> > In line 138 the author indicates that the smaller value the higher priority is, but in line 140 it indicates the algorithm schedules the largest value, the description here is confusing.
>
> Thanks very much for pointing out the typo, the `max` in line 140 should be `min`. We have revised the typo in the updated manuscript.
>
> > The description of the encoding operator in line 181 is not clear, and further normalized features can reduce the impact of noises, which is not well explained in the paper.
>
> The ADD3 in line 181 is encoded as [0, 1, 1, 0.5, 0.5] in the format of [is_mul, is_add, operations' number of cycles, normalized ASAP, normalized ALAP]. We encode the operation type using one-hot encoding such that add is encoded as [0, 1] and mul is encoded as [1, 0]. If there are more types of operations, the one-hot encoding could be extended similarly. And the number of cycles for add is 1.
>
> The normalization of ASAP and ALAP is conducted to reduce the impact of noises brought by different depths of different CDFGs. For instance, an operation node with ASAP value as k means quite different in a k-layer CDFG against a 2k-layer CDFG, which should be distinguished. So we normalize the ASAP and ALAP values with the number of layers of  a CDFG.
>
> > In Section 5.2, it is mentioned that V_chosen on line 25 of the algorithm, but the algorithm has only 23 lines, which makes it difficult to understand.
>
> Thanks very much for pointing out the typo! Here, line 25 should be line 20, and line 22 should be line 16, and line 10 should be line 7. We have updated the algorithm description in the newly-uploaded manuscript.

---

> ### Author Response · Authors · 2022-08-02
> **Authors' Response to Reviewer PRGS**
>
> We thank the reviewer for your valuable comments and respond to the questions below.
>
> > What is the final time interval for different scheduling sequences in Figure 5 (b) and (c)? This supplement would give a intuitive explanation on the operation's flexibility during scheduling.
>
> In the example shown in Figure 5 (b) and (c), whether considering the operation's flexibility or not would not result in  different scheduling sequences. However, considering a slightly more complicated example: If we add an edge (0, 1), which means that MUL1 must be executed after ADD0, the ASAP result will be as follows.
>
> |        | ADD  |  MUL  |
> | :----: | :----: | :----: |
> | Step 0 | ADD0, ADD2 | |
> | Step 1 | | MUL1 |
> | Step 2 | ADD3 | MUL4 |
> | Step 3 | ADD5 | |
>
> And the ALAP result will be:
>
> |        | ADD  |  MUL  |
> | :----: | :----: | :----: |
> | Step 0 | ADD0 | |
> | Step 1 | | MUL1 |
> | Step 2 | ADD3, ADD2 | |
> | Step 3 | ADD5 | MUL4 |
>
> The operations' flexibilities are:
> | ADD0 |  MUL1 | ADD2 | ADD3 | MUL4 | ADD5 |
> | :----: | :----: | :----: | :----: | :----: | :----: |
> | 0 | 0 | 2 | 0 | 1 | 0 |
>
> The flexibility of ADD2 is higher than ADD0. Suppose that both ADD and MUL take one cycle. With one adder and one multiplier, if ADD2 is scheduled before ADD0, the final scheduling order would be:
>
> |        | ADD  |  MUL  |
> | :----: | :----: | :----: |
> | Cycle 0 | ADD2 |      |
> | Cycle 1 | ADD0 |      |
> | Cycle 2 |      | MUL1 |
> | Cycle 3 | ADD3 | MUL4 |
> | Cycle 4 | ADD5 |      |
>
> From the above table, the execution of the CDFG takes 5 cycles. Moreover, if we consider the node flexibility, ADD0 should be scheduled before ADD2. The final scheduling order would be:
>
> |        | ADD  |  MUL  |
> | :----: | :----: | :----: |
> | Cycle 0 | ADD0 |      |
> | Cycle 1 | ADD2 | MUL1 |
> | Cycle 2 | ADD3 | MUL4 |
> | Cycle 3 | ADD5 |      |
>
> Now the execution of the CDFG takes 4 cycles. Therefore, the scheduling sequence considering the operation's flexibility has a better result.

---

### Official Review · Reviewer_2hWG · 2022-07-13

**Rating:** 7
**Confidence:** 4
**Soundness:** 3 good
**Presentation:** 3 good
**Contribution:** 3 good

**Summary:**

The authors propose NeuroSchedule, a GNN-based schedule for solving the high-level synthesis (HLS) problem in the FPGA development. NeuroSchedule is the first GNN-based scheduler in this domain. It is trained using supervised learning with data labeled by the ILP-based scheduler. NeuroSchedule has an extremely good runtime improvement over the ILP-based scheduler, while achieving near-optimal solutions. NeuroSchedule is also compared with a heuristic-based method. Its run-time is not as good as the heuristic method, but its solutions are generally better.

**Questions:**

-	Should “max” in Line 41 be “min”, since the list scheduling algorithm picks the operations with the highest priority first and smaller F values indicate higher priorities?

**Limitations:**

The experiment shows the proposed method is slower than the heuristic entropy-directed method.

It would be better to mention how long it took to label the training data set of 50,000 CDFGs by the ILP-based scheduler. Not necessary a limitation, but the readers may want to know.



**Strengths And Weaknesses:**

Strengths:

-	The inference time of the proposed algorithm is 50,000 times faster than its teacher schedule (ILP-based optimizer).
-	The quality of solutions provided by NeuroSchedule are better than the state-of-the-art entropy-directed method by 6.10% on average.
-	By fine-tuning the pre-trained model, NeuroSchedule can get good results in different settings.
-	The rank-loss (relative priorities) helps the trained model to get a better optimal rate, compared with traditional MSE loss.
-	The experiment section is well organized.

Weaknesses:

-	It would be better to conduct significant test to see whether the performance of NeuroSchedule is significantly better than the entropy-directed method.
-	It would be better to compare with some reinforcement learning based scheduling methods.
-	There may be a typo in one of the formulas.

---

> ### Author Response · Authors · 2022-08-02
> **Authors' Response to Reviewer 2hWG**
>
> We thank the reviewer for your valuable comments and respond to the questions below.
>
> > Should “max” in Line 41 be “min”, since the list scheduling algorithm picks the operations with the highest priority first and smaller F values indicate higher priorities?
>
> Thanks very for pointing out the typo. It should be `min` instead of `max`.  We have corrected the typo in the updated manuscript.
>
> > It would be better to mention how long it took to label the training data set of 50,000 CDFGs by the ILP-based scheduler. Not necessary a limitation, but the readers may want to know.
>
> It takes about 15 hours to label the training dataset of 50,000 CDFGs by the ILP-based scheduler. We use Gurobi as the ILP-solver and the CPU is (Xeon(R) Gold 6271C CPU@2.60GHz).

---

### Official Review · Reviewer_1vbZ · 2022-07-20

**Rating:** 5
**Confidence:** 2
**Soundness:** 3 good
**Presentation:** 4 excellent
**Contribution:** 3 good

**Summary:**

This paper proposes NeuroSchedule, an efficient and effective GNN-based scheduling method designed for the HLS. This paper formulate the learning problem of the operation scheduling in HLS and is able to embed the operation information in the GNN to aid learning the schedule. This paper further builds a dataset of CDFGs with different scheduling settings to pre-train the model. With final-tuning on randomly generated designs in different settings, the model's prediction is able to be generalized.

**Questions:**

1. HLS seems to be not that related in this article. This article seems more about the general operation scheduling. Is there any special reason or limitation why the authors decide the scope of HLS?
2. How do the authors solve the case of resource conflict, if the hardware modules are to be reused? Not only the operation dependency will influence the scheduling, whether the hardware module busy or not will also have impact. For instance in Fig5.a, what if there are some restrictions that the first two addition have to use the same adder
3. Will and how do the authors consider HLS's different pargmas in the scheduling, as they will decently impact the scheduled performance via changing the underlying hardware, such as buffer memory size/bandwidths?


**Limitations:**

Yes, it's addressed.

**Strengths And Weaknesses:**

Strength:
1. This paper proposes a formulated and more likely to be standardized method to solve the seemingly intractable scheduling problem in HLS. This may open door and inspire more ML techniques to be effectively applied in the hardware design.
2. The methodology description along with the supplementary material is clear and easy for the readers to follow or potentially reproduce

Weakness:
1. The results' improvements are hard to be justified as significant, compromising the method's effectiveness: As shown in Table 1, as compared with Entropy-directed methods, overall the runtime latency is improved by 16.67% but the optimization run-time is 5.7 times more. Further compare with the eligible options in ILP-based, it is shown that Entropy-directed method is already near optimal. Then it is questionable if it is worth it for the relatively small latency improvement but with huge degradation on the scheduling run-time.
2. A little too simplified scheduling assumption based mainly on individual operation latency and dependency, but authors can clarify in the following questions to clear the concern.

---

> ### Author Response · Authors · 2022-08-02
> **Authors' Response to Reviewer 1vbZ**
>
> We thank the reviewer for your valuable comments and respond to the questions below.
>
> > HLS seems to be not that related in this article. This article seems more about the general operation scheduling. Is there any special reason or limitation why the authors decide the scope of HLS?
>
> We are focusing on the study of HLS methods. As depicted in Figure 1, the HLS flow is composed of compilation, allocation, scheduling, binding and generation. The scheduling phase in HLS is a special kind of operation scheduling  problem, which focuses on operations bounded in a graph structure (i.e. CDFG). We agree that the addressing scheduling problem is similar to the general operation scheduling problem. But due to the page limit, we did not compare our proposed method with state-of-the-art scheduling methods on general benchmarks. However, we believe our proposed method could be extended to tackle the general operation scheduling problems. We will conduct the comparison in our future work.
>
> > How do the authors solve the case of resource conflict, if the hardware modules are to be reused? Not only the operation dependency will influence the scheduling, whether the hardware module busy or not will also have impact. For instance in Fig5.a, what if there are some restrictions that the first two addition have to use the same adder?
>
> The resource conflicts problem is solved in our NeuroSchedule algorithm. As shown in lines 8 and 13 of Algorithm 1, the available functional units $u_j$ (i.e. hardware modules) are allocated to ready operations respectively according to their priority. Moreover, in lines 13 and 23, busy functional units in current time step are skipped to avoid the resource conflicts. In Fig5.a, if ADD0 and ADD2 have to use the same adder, our NeuroSchedule would first schedule ADD0 to use the adder since ADD0 has higher priority than ADD2. After ADD0 is finished, ADD2 is scheduled to used the adder to avoid adder conflict.
>
> > Will and how do the authors consider HLS's different pargmas in the scheduling, as they will decently impact the scheduled performance via changing the underlying hardware, such as buffer memory size/bandwidths?
>
> As depicted in Figure 1, the HLS flow are composed of compilation, allocation, scheduling, binding and generation.  Different pragmas are considered in different phases of HLS. For example, loop optimizations, such as loop tiling and loop unrolling, are processed in compilation phase to generate high-performance CDFGs for scheduling phase. Buffer memory size and bandwidths configuration are considered in allocation phase for choosing desirable hardware resources. In this paper, we focus on the scheduling problem.

---

### Official Review · Reviewer_tUpk · 2022-07-27

**Rating:** 6
**Confidence:** 5
**Soundness:** 3 good
**Presentation:** 4 excellent
**Contribution:** 3 good

**Summary:**

This paper proposes NeuroSchedule, a GNN-based scheduling method for HLS. At the core of NeuroSchedule is a novel priority function that predicts the priority (rank) of a ready operation, which is used for list scheduling. To improve the scalability of NeuroSchedule over different settings, pre-training methods are adopted, which can be fine-tuned for downstream tasks. NeuroSchedule effectively approximates the optimal ILP-based priority function at a 50,000x lower computation cost. This leads to an average of 6.10% improvement in cycles over the Entropy-Directed Scheduling (EDS) algorithm.

**Questions:**

* How would NeuroSchedule handle more complex/realistic applications with non-deterministic events such as DRAM accesses, caching, bank conflicts, etc?

* In Equation (2), why ALAP includes the second definition with ASAP() for those vertices not connected to $v_i$? Can you elaborate on this?

**Limitations:**

I find this paper presents an interesting proposal and was a very enjoyable read. As far as I know, this is the first work that formulates the scheduling priority function as a learning task with GNN to effectively approximate the optimal ILP algorithm at a tiny fraction of computation cost. Indeed, it's a promising direction and the results demonstrate the effectiveness of the proposed idea.

While I like the direction of this work, my acceptance is *weak* for the following limitations. For benchmark selection, most of the benchmarks look pretty small in scale, taking tens of cycles in most cases. At this point, it is not clear how effective NeuroSchedule would be for more realistic/complex applications that may require off-chip DRAM accesses, caches, and other non-deterministic behaviors. Also, while delivering competitive results to the optimal ILP-based schedule in most cases, the average improvement over EDS is rather small. This makes me question about the practical value of NeuroSchedule as its average improvement over EDS would go below 5% if we include those optimal benchmarks omitted while it takes about 6$\times$ longer scheduling time.

**Strengths And Weaknesses:**

**Strengths**

* The approach of applying GNN to HLS scheduling is novel and clever.

* NeuroSchedule demonstrates competitive results to the optimal ILP algorithm at a tiny faction of computation cost.


**Weaknesses**

* The effectiveness of NeuroSchedule is demonstrated only for small-scale kernel benchmarks taking just tens of cycles in most cases.

* The relatively small average improvement over EDS (i.e., 6.10% *without* including those with 0.0% improvement) while taking some 6$\times$ longer to run.

---

> ### Author Response · Authors · 2022-08-02
> **Authors' Response to Reviewer tUpk**
>
> We thank the reviewer for the valuable comments and respond to the questions below.
>
> > How would NeuroSchedule handle more complex/realistic applications with non-deterministic events such as DRAM accesses, caching, bank conflicts, etc?
>
> NeuroSchedule handles memory operations (e.g. load and store) using on-chip memories like BRAMs, for which the memory access time is deterministic. The non-deterministic events are not considered in NeuroSchedule. However, we believe non-deterministic events could be effectively addressed by extending our proposed method. Moreover, due to lack of commonly used large-scale training data (ILP-based method cannot label large-scale benchmarks), NeuroSchedule currently only works on small-scale benchmarks. We are working on alternative labeling methods for large-scale training data, which will extend the capacity of NeuroSchedule on large-scale cases. We have also stated this extension in our future work in the manuscript.  And we will include non-deterministic events as our future topic.
>
> > In Equation (2), why ALAP includes the second definition with ASAP() for those vertices not connected to vi? Can you elaborate on this?
>
> In our method, ASAP is computed before ALAP to obtain the depth of CDFG $G$ (i.e. $\max \limits_{v_k \in V} ASAP(v_k)$ ). In Equation (2), for those vertices not connected to $v_i$, their ALAP values are defined as the depth of CDFG G (i.e. $\max \limits_{v_k \in V} ASAP(v_k)$ ), according to the ALAP algorithm. Naturally, the operations at the bottom of the CDFG are scheduled as-late-as-possible.

---

### Author Response · Authors · 2022-08-02
**Authors' Response to Reviewers**

We thank the reviewer for the valuable comments. We noticed that the reviewers are concerned about the scheduling results and the runtime of our method compared with EDS. First, for HLS, 6.1% improvement in the scheduling result is significant enhancement, which notably reduces the runtime of the circuits and thus enhances the circuit's performance. Second, in the whole HLS flow, the scheduling process only takes a small fraction of the total runtime, and therefore the increased runtime does not notably slow down the HLS process. Third, one circuit design corresponds to millions of hardware implementations. Considering the small burden in HLS runtime (only executed once), it is easy to understand that the designers are much more concerned with the circuit's performance. Therefore, considering the significant enhancements in scheduling results, the increased runtime is acceptable. Moreover, we will include accelerating NeuroSchedule as our future work.

---

### Meta-Review · Area_Chair_AbpX · 2022-08-26

**Recommendation:** Accept
**Confidence:** Certain

**Metareview:**

This paper presents NeuroSchedule, a GNN-based scheduling method for high-level synthesis. To enhance the scalability for various scheduling problems with different settings, the paper additionally adopts pre-training/fine-tuning methodology. For pre-training, the paper use data labeled by the ILP-based scheduler in supervised way fashion. Experimental results indicate that NeuroSchedule can obtain near-optimal solution with significant speedup compared with traditional method.

All reviews have similar opinion on the strength and weakness of the paper. On the strength side, the paper is well written, and details are clearly explained. Both the GNN-based scheduling method and pre-training/fine-tuning approach are novel and interesting. The experiments are reasonably sound.

The concerns are 1). The testing benchmarks are small; and 2). The optimization run time is longer than the known Entropy-Directed scheduling method. During rebuttal period, the authors provide further explanation, partially addressing the concerns.

Overall, I think it is a solid paper, particularly in the aspect of novelty. Therefore, the paper is recommended for acceptance.


**Award:**

No

---

### Decision · Program_Chairs · 2022-09-14

Accept